# Archaeal chromatin 'slinkies' are inherently dynamic complexes with deflected DNA wrapping pathways

Samuel Bowerman[1], Jeff Wereszczynski[2], Karolin Luger[1]*

[1]Department of Biochemistry and Howard Hughes Medical Institute, University of Colorado Boulder, Boulder, United States; [2]Department of Physics and Center for the Molecular Study of Condensed Soft Matter, Illinois Institute of Technology, Chicago, United States

**Abstract** Eukaryotes and many archaea package their DNA with histones. While the four eukaryotic histones wrap ~147 DNA base pairs into nucleosomes, archaeal histones form 'nucleosome-like' complexes that continuously wind between 60 and 500 base pairs of DNA ('archaeasomes'), suggested by crystal contacts and analysis of cellular chromatin. Solution structures of large archaeasomes (>90 DNA base pairs) have never been directly observed. Here, we utilize molecular dynamics simulations, analytical ultracentrifugation, and cryoEM to structurally characterize the solution state of archaeasomes on longer DNA. Simulations reveal dynamics of increased accessibility without disruption of DNA-binding or tetramerization interfaces. $Mg^{2+}$ concentration influences compaction, and cryoEM densities illustrate that DNA is wrapped in consecutive substates arranged 90° out-of-plane with one another. Without ATP-dependent remodelers, archaea may leverage these inherent dynamics to balance chromatin packing and accessibility.

*For correspondence:
karolin.luger@colorado.edu

Competing interests: The authors declare that no competing interests exist.

## Introduction

Eukaryotic genomes are orders of magnitude larger and more complex than those of archaea or bacteria. They manage their massive genomes through a hierarchical packaging scheme that utilizes histones to form nucleosomes, a complex that contains two H2A-H2B histone heterodimers flanking a central (H3-H4)$_2$ heterotetramer and stably wraps ~147 base pairs of DNA (*Figure 1A*; *Luger et al., 1997*). While all eukaryotes invariably contain histones, genes encoding putative 'minimalist' histone proteins have been identified in most archaea (*Henneman et al., 2018*), but so far not in bacteria. While still a subject of debate, the presence of histones in the two domains of life has supported theories that eukaryotes evolved directly from an archaeon (*Heinicke et al., 2004*; *Malik and Henikoff, 2003*; *Brunk and Martin, 2019*; *Watson, 2019*; *Eme et al., 2017*; *Spang et al., 2018*; *Sandman and Reeve, 2006*).

Archaeal histones share many features with their eukaryotic equivalents, such as the three-helix histone fold motif (α1-L1-α2-L2-α3) (*Decanniere et al., 2000*), as well as obligate dimerization and transient tetramer formation (*Marc et al., 2002*). In both archaea and eukaryotes, histones have a preference for occupying particular sites in the genome (*Ammar et al., 2012*; *Nalabothula et al., 2013*). Eukaryotic histones utilize highly disordered cationic N-terminal tails to regulate gene transcription and chromosome compaction via post-translational modifications (PTMs) (*Luger and Richmond, 1998*; *Musselman et al., 2012*; *Bowman and Poirier, 2015*), whereas most archaeal histones do not contain tail sequences, with the exception of some sequences found within the Asgard clade (*Henneman et al., 2018*). Additionally, histones in eukaryotes have evolved histone fold 'extensions', unique secondary structure elements that define the outer surface of the nucleosome as well as

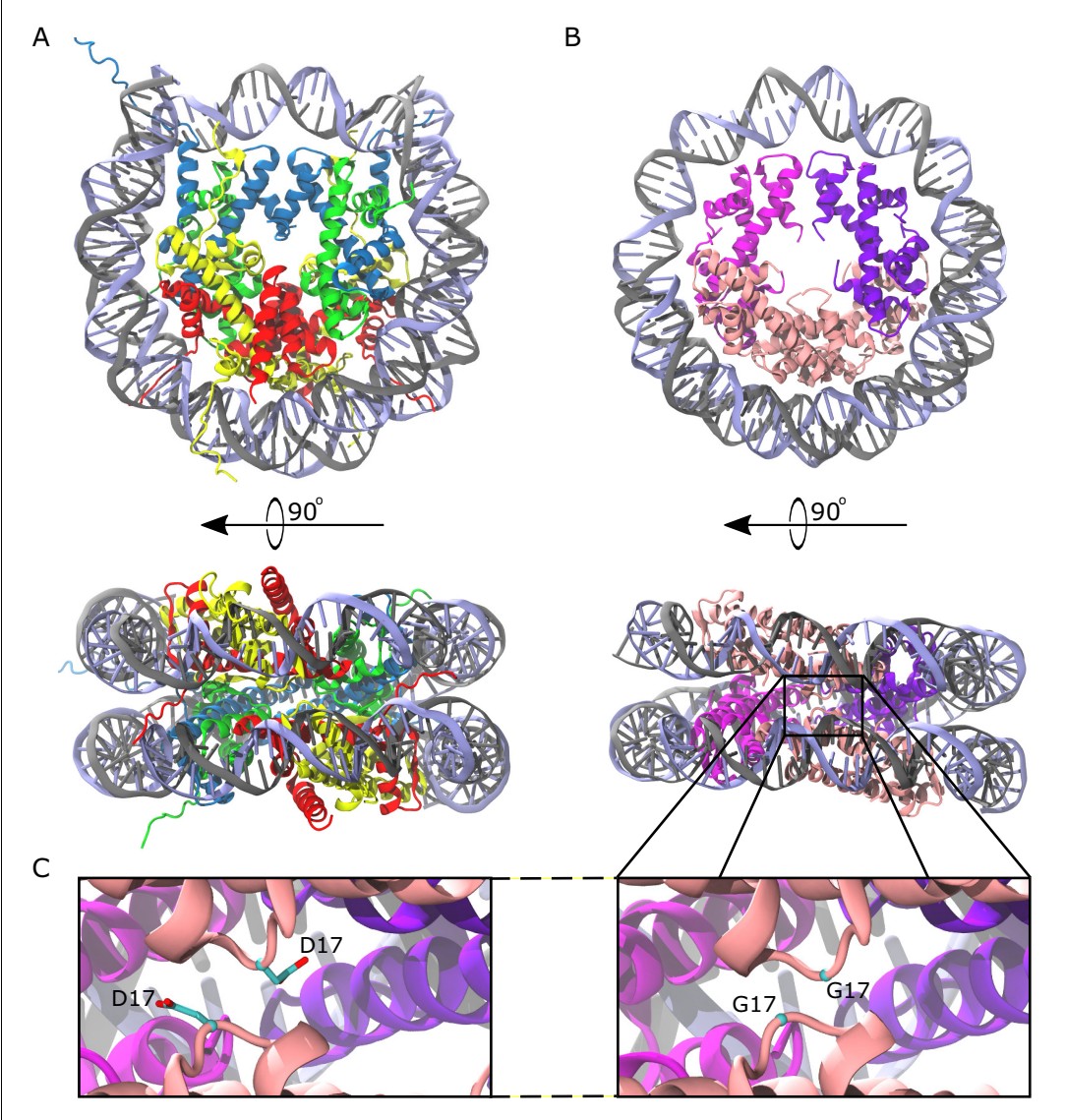

**Figure 1.** Comparison of the eukaryotic nucleosome and Arc120, an archaeasome model system containing 120 base pairs of DNA and four histone dimers. (**A**) The eukaryotic nucleosome (PDB 1AOI) containing two copies of H3, H4, H2A, and H2B (blue, green, gold, and red, respectively) arranged as H2A-H2B dimers flanking the (H3–H4)$_2$ tetramer and binding 147 base pairs of DNA (gray and light blue). (**B**) The model Arc120 system, derived from the Arc90 crystal structure (PDB 5T5K), with four HMfB homodimers participating in L1-L1 stacking interactions shown in pink. (**C**) Enhanced views of both the wild type and G17D mutant interfaces simulated in this study.

stabilize the histone core and the final turn of nucleosomal DNA. No such diversification has yet been identified in any of the archaeal histones.

Recently, we determined the crystal structure of *Methanothermus fervidus* histone HMfB bound to ~90 base pairs of DNA (*Mattiroli et al., 2017*; *Bhattacharyya et al., 2018*). HMfB is a typical representative of most archaeal histones, as it contains no tails and no histone fold extensions, and it exists as either a homodimer or heterodimer with the closely related HMfA histone (*Sandman et al., 1994*). Nevertheless, striking similarities to the eukaryotic nucleosome were observed, including a superhelical DNA path around a histone core that is nearly identical to that in nucleosomes, promoted by conserved DNA contacts with histones, and by dimer-dimer interactions through 'four-helix bundles' of α-helices. However, unlike nucleosomes, which are defined octameric particles that arrange as 10 nm 'beads on a string' on long DNA fragments, crystal contacts in the archaeal system promote an extended superhelical winding of DNA around a histone core that consists of more than

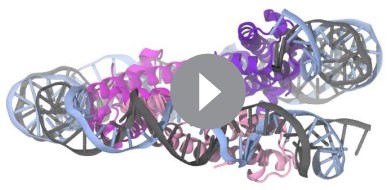

**Video 1.** Rotating view of the Arc90 system. This system contains three histone dimers and no stacking interactions.

https://elifesciences.org/articles/65587#video1

the canonical four histone dimers. This arrangement, which is consistent with in vivo footprinting results (*Mattiroli et al., 2017*), places the L1 loops of every $n^{th}$ and $(n+3)^{th}$ dimer in close proximity to one another (*Figure 1C*). The importance of the L1-L1 interaction in stabilizing this arrangement in vitro and in vivo was established by substituting the highly conserved G17 residue with bulkier amino acids (e.g. G17D), resulting in transcription regulation defects.

Here, we show that archaeal histones, when assembled on long segments of DNA (>200 base pairs), experience disparate solution dynamics compared to eukaryotic nucleosomes, and we present cryo-EM structures of two classes of particles arranged on a 207 base pair DNA fragment with deflections of the wrapping pathway. Despite many nucleosome-like properties, we refer to these constructs as 'archaeasomes', rather than 'archaeal nucleosomes', because of their inherent differences in higher order structures and solution behaviors, including the ability to form histone cores with more than four dimers. We classify archaeasomes by the length of bound DNA (i.e. shorthand of 'Arc120' for 'archaeasome with 120 base pairs of DNA'). We use molecular dynamics (MD) simulations to show that archaeasomes can expand, like the stretching of a 'slinky', to increase overall accessibility without sacrificing histone-histone or histone-DNA interactions. Simulations also reinforce the importance of L1-L1 histone stacking interactions in regulating this dynamic equilibrium. Sedimentation velocity analytical ultracentrifugation (SV-AUC) and cryoEM highlight two distinct accessibility dynamics at play in archaeasome systems, and cryoEM analysis identifies archaeasomal states that continually wrap DNA but deflect the wrapping pathway 90° out-of-plane. The intrinsic dynamic behavior of archaeasome slinkies stands in contrast to eukaryotic nucleosomes, which are compact and highly stable and require modifications and elaborate machinery to efficiently regulate chromatin accessibility, whereas archaea may utilize this inherent stochastic behavior in order to access their DNA.

## Results

### Archaeasomes are highly dynamic but maintain robust dimer-dimer and dimer-DNA interactions in simulations

We applied molecular dynamics (MD) simulations to model the stability and solution dynamics of archaeasomes assembled on DNA of increasing lengths. Three system sizes were studied: an archaeasome with 90 base pairs of DNA ('Arc90', *Video 1*), one with 120 base pairs of DNA ('Arc120', *Video 2*), and one with 180 base pairs of DNA ('Arc180', *Video 3*). In *T. kodakarensis*, the G17D (G16D in HMfB) mutation in the L1 loop was previously shown to abolish the characteristic footprint of histone-based chromatin in the cell (*Mattiroli et al., 2017*), and we simulated this system at the archaeasome level ('Arc120-

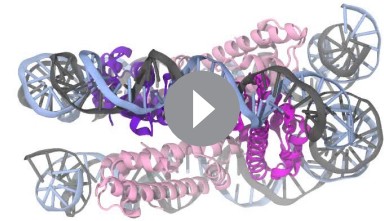

**Video 2.** Rotating view of the Arc120 system. This system contains four histone dimers and one L1-L1 stacking interaction. Both the wild-type and G17D mutant of this system were simulated.

https://elifesciences.org/articles/65587#video2

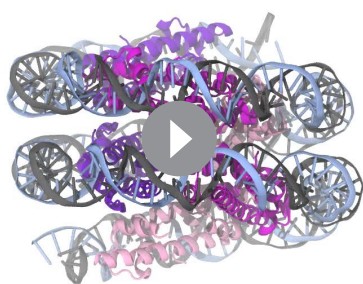

**Video 3.** Rotating view of the Arc180 system. This system contains six histone dimers and three L1-L1 stacking interactions.
https://elifesciences.org/articles/65587#video3

G17D') to elucidate the atomistic mechanisms for this observed behavior.

In each system, the DNA footprint of ~30 base pairs per dimer was maintained on the hundreds of nanoseconds timescale. Nevertheless, the Arc90 system experienced a high degree of global dynamics. RMSD calculations of backbone atom positions showed a maximum deviation of ~6 Å from the initial Arc90 coordinates, with trough-to-peak heights of ~4 Å throughout the time course (*Figure 2A,B*). RMSF calculations identify regions of highest flexibility where the DNA binds to the terminal dimers (*Figure 2—figure supplement 1*), and visual inspection of Arc90 trajectories revealed that the system samples a 'clamshell' motion, where terminal dimers and associated DNA fluctuate to create 'closed' and 'open' accessibility states (*Videos 4* and *5*). The equilibrium between closed and open forms was quantitated by measuring the center-of-mass distance between each DNA end and the neighboring superhelical turn of DNA, where separations of ~16 Å and ~30 Å are indicative of closed and open states, respectively (*Figure 2C*, *Figure 2—figure supplement 2*). Over the clamshell motion, the four-helix bundle interfaces that bind consecutive dimers to one another was maintained, and only small rearrangements of the local contact network were observed. Together, these data show that archaeasomes retain protein-protein contacts while still allowing for significant molecular motion.

Simulations of Arc120 and Arc180 systems showed that L1-L1 interactions drastically reduced the population of the accessible state. Both the maximum observed RMSD and variance in the timeseries were dampened in these systems (*Figure 2A,B*), and RMSF calculations showed that DNA bound to the terminal dimers were less dynamic than in Arc90 trajectories (*Figure 2—figure supplement 1*). Similarly, the distance between DNA entry and exit sites and the neighboring DNA gyre in Arc120 and Arc180 were unimodal around closed-state values (*Figure 2C*), with average separations of $17.4 \pm 0.2$ Å and $16.1 \pm 0.2$ Å, respectively.

While the Arc120-G17D system has the proper number of dimers to form L1-L1 stacking interactions, the G17D mutation was designed to disfavor interactions, and this system experienced increased dynamics relative to the wild-type Arc120 trajectories. RMSD measurements showed a unimodal distribution, as was seen in the Arc120 and Arc180 simulations, but the maximum observed RMSD values were more similar to the Arc90 trajectories (*Figure 2*). Similarly, local fluctuations in Arc120-G17D DNA positions were increased relative to the Arc120 trajectories (*Figure 2—figure supplement 1*), and separation of the superhelical gyres was significantly larger than the Arc120 values ($21.0 \pm 0.4$ Å, p-value<0.0001). This also increased the solvent-accessible surface area by over 1,000 Å$^2$ ($57,316 \pm 34$ Å$^2$ vs $58,422 \pm 73$ Å$^2$, p-value<0.001; *Figure 3*). Similar to the Arc90 system, the four-helix bundle interactions of consecutive dimers were maintained in these systems, despite the increase in DNA separation and surface accessibility.

Histone-DNA interactions were largely unchanged by these dynamics. Molecular Mechanics-Generalized Born Surface Area (MM-GBSA) calculations were used to estimate histone-DNA binding energies (*Table 1*). MM-GBSA values should not be interpreted literally, but can be used to identify qualitative and robust changes in DNA binding interactions in nucleosomes as a result of histone modifications (*Bowerman and Wereszczynski, 2016*; *Bowerman et al., 2019*; *Morrison et al., 2018*). In agreement with previous computational studies (*Rojec et al., 2019*), raw MM-GBSA values suggest that DNA-binding strength increases as additional dimers are added to an archaeasome stack ($\Delta G_{Arc90} < \Delta G_{Arc120} < \Delta G_{Arc180}$). However, this may be an artifact of molecular mechanics force-fields, which are additive and thereby affected by the total number of interactions in a system. When MM-GBSA values were normalized to the number of dimers in each system, there was no net change in DNA binding ability as a function of archaeasome size. Additionally, the G17D point mutation

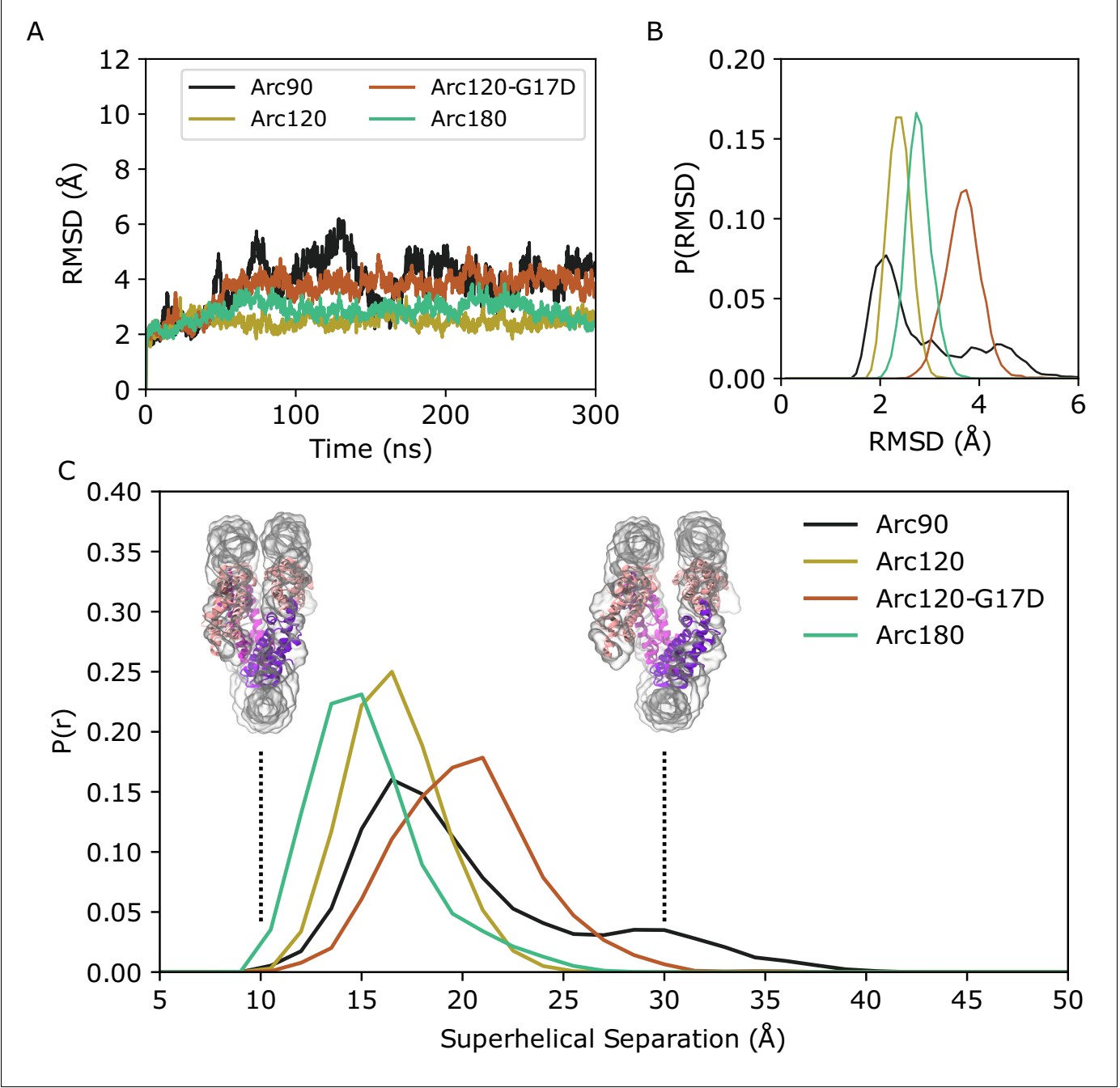

**Figure 2.** Backbone RMSD for simulated systems compared to their initial conformations. (**A**) Representative timeseries from each system demonstrating the largest change in structure from the initial states. The Arc90 trajectory (black) exhibits both a large peak value, as well as wide variances across the timeseries. Arc120 (gold) and Arc180 (teal) systems are significantly less dynamic, and the Arc120-G17D mutant (orange) shows an increased divergence from the initial state when compared to the wild-type Arc120 system. (**B**) Distribution of RMSD values sampled across all three independent trajectories of each system, post-equilibration (100 ns). Arc90 displays a sampling of two different states, while each of the larger systems are unimodal. (**C**) Distribution curves for center-of-mass separations between DNA ends and the neighboring superhelical gyre. Representative conformations of fully closed (~10 Å) and open (~30 Å) states identified in the Arc120-G17D simulations are shown. Arc90 simulations (black) show a bimodal distribution between these values, but Arc120 (gold) and Arc180 (teal) systems, containing one and three L1-L1 interactions each, sample unimodal values around the closed state. The Arc120-G17D mutant (orange) samples a unimodal distribution, but indicative of a more open archaeasome than the wild type.

The online version of this article includes the following source data and figure supplement(s) for figure 2:

**Source data 1.** RMSD timeseries data for replicates of molecular dynamics trajectories for *Figure 2A,B*.

*Figure 2 continued on next page*

*Figure 2 continued*

**Figure supplement 1.** Root mean-squared fluctuation (RMSF) plots for measured for DNA bases in all four simulated systems.
**Figure supplement 1—source data 1.** RMSF vs residue plots for replicate trajectories of simulated systems for *Figure 2—figure supplement 1*.
**Figure supplement 2.** Representative timeseries for DNA end-to-neighbor separation distances from each system.

slightly reduced the dimer-DNA interaction strength, but this difference was not statistically significant when compared to the wild-type Arc120 system (p-value of 0.157). These data, in conjunction with maintained four-helix bundle interactions, show that archaeasomes can sample open and accessible conformations without histone dissociation. Thus, the loss of the 30 base pair ladder pattern in MNase digestion of chromatin isolated from cells carrying the G17D mutation as their only source of histones may be due to increased archaeasome dynamics (*Figure 3*), rather than frequent histone dissociation (*Mattiroli et al., 2017*).

## Archaeasomes are inherently dynamic and accessible in solution

Our simulations provide in silico confirmation for the notion that archaeasomes indeed wrap several DNA turns, with more than four histone dimers, as a bona fide solution-state. The Arc90 and Arc120-G17D simulations suggest that archaeasomes may sample extended conformations without even partial dissociation of histones from either DNA or their histone partners. To structurally characterize these assemblies in solution, we reconstituted archaeasomes on 207 base pairs of Widom 601 DNA (Arc207) with saturating amounts of recombinant HTkA histones from *Thermococcus kodakarensis*, which are very similar to HMfB histones (overall 59% identity, 81% similarity, with higher conservation in sequences involved in histone-histone and histone-DNA interactions) (*Mattiroli et al., 2017*). The solution behavior of the Arc207 construct was then analyzed by single particle cryoEM and sedimentation velocity analytical ultracentrifugation (SV-AUC). As controls, we also collected SV-AUC traces from eukaryotic (*Xenopus laevis*) nucleosomes reconstituted on 147 base pairs of Widom 601 DNA (*Lowary and Widom, 1998*).

Gel shift assays showed that full complex saturation occurs when DNA and histones are mixed at the previously reported stoichiometric limit (~30 bp per dimer; *Figure 4—figure supplement 1*; *Mattiroli et al., 2017*). SV-AUC traces show that the Arc207 complex sediments homogeneously at ~10.6 s (*Figure 4*, circles), and the Nuc147 control sediments more rapidly than the Arc207 system at ~11.2 s (*Figure 4*, squares). As sedimentation is dependent on both molar mass (higher mass yields faster sedimentation) and shape anisotropy (higher anisotropy yields increased drag and slower sedimentation), the relatively lower sedimentation rate of the Arc207 sample has one of two

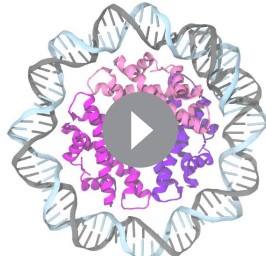

**Video 4.** Face-view of dynamics observed in a single Arc90 trajectory. While the terminal dimers exhibit large fluctuations around the central histones, four-helix bundle interactions are maintained.
https://elifesciences.org/articles/65587#video4

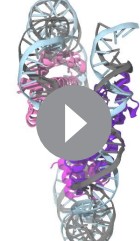

**Video 5.** Side-view of dynamics observed in a single Arc90 trajectory. The dynamics of the terminal dimers creates 'open' and 'closed' states in a 'clamshell-like' motion.
https://elifesciences.org/articles/65587#video5

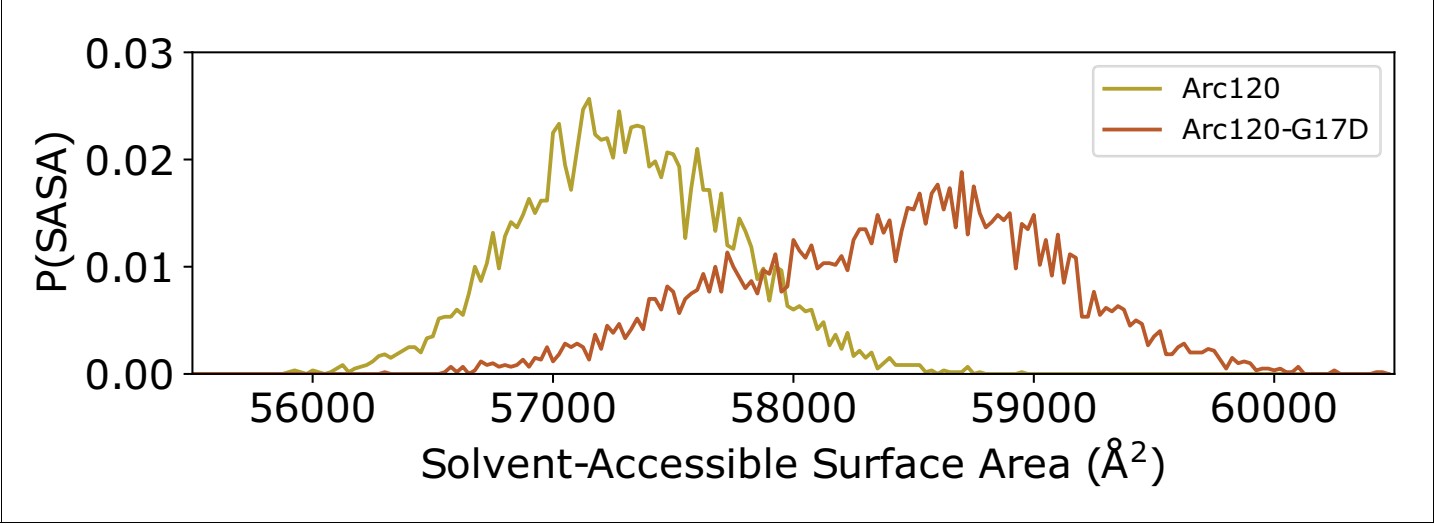

**Figure 3.** Distribution of solvent-accessible surface areas observed per frame in the Arc120 and Arc120-G17D simulations. The G17D mutation at the key L1-L1 interaction between stacking dimers significantly increases the surface area by ~1000 Å$^2$.

The online version of this article includes the following source data for figure 3:

**Source data 1.** Timeseries of solvent-exposed surface area in Arc120 and Arc120-G17D simulations for *Figure 3*.

interpretations: either its mass is less than that of a nucleosome, or the particle is more extended in solution, exhibiting higher shape anisotropy and drag.

The mass and anisotropy of each molecule were modeled from SV-AUC traces using the Monte-Carlo-coupled Genetic Algorithm (GA-MC) module of Ultrascan III (*Table 2*). The molecular weight for both particles was slightly overestimated, consistent with a systematic error in specific volume estimation for protein-DNA complexes, which can propagate to modest errors in molecular weight estimation (*Demeler et al., 2014*). Nevertheless, these data show that the Arc207 complex is fully saturated with histones, that the total mass is consistent with seven histone dimers assembled on 207 bp of DNA, and that its mass is greater than that of Nuc147. This suggests that the Arc207 system is more anisotropic than Nuc147 in order to satisfy the decrease in sedimentation rate, which is confirmed by the frictional ratios ($f/f_o$) estimated from GA-MC calculations (1.94 and 1.55 for Arc207 and Nuc147, respectively). For reference, the $f/f_o$ value of free 207 DNA is 3.14 so the Arc207 complex is more compact than free DNA but not as compact as the eukaryotic nucleosome.

We next utilized single particle cryoEM to capture the three-dimensional structure of the Arc207 complex. Our simulations of the similarly sized Arc180 system predicted that the complex would be compact, but SV-AUC modeling conversely suggests that Arc207 forms an extended conformation. CryoEM data shows that the Arc207 system is indeed open, as it exhibits nucleosome-like particle dimensions with increased spacing between neighboring DNA gyres (*Figure 4B*), in agreement with SV-AUC data. This is also apparent in the two-dimensional classifications (*Figure 4C*). Because of the apparent structural heterogeneity of this assembly, we were unable to extract three-dimensional conformations from the dataset, even at low resolution.

## Archaeasomes compact but do not oligomerize in the presence of Mg$^{2+}$

Eukaryotic chromatin fibers can be compacted in vitro through the addition of divalent cations such as Mg$^{2+}$ (*Schwarz et al., 1996*). To test whether this also applies to archaeal chromatin, we analyzed Arc207 samples in the presence of 0, 1, 2, 5, 7, 8, and 10 mM MgCl$_2$ by SV-AUC, and we observed changes in both sedimentation coefficient and frictional ratio (*Figure 5*). Arc207 samples exhibit an increase in sedimentation rate with increased Mg$^{2+}$ concentration from 0 to 5 mM, with no additional increase between 5 and 10 mM MgCl$_2$ (*Figure 5A*, top). In comparison, the nucleosome system shows little to no change in sedimentation from 0 to 2 mM MgCl$_2$ (*Figure 5A*, bottom), and start to self-associate (aggregate) at concentrations above 2 mM Mg$^{2+}$.

**Table 1.** MM-GBSA calculations for DNA binding strength, both as raw values and normalized to the number of dimers in each system.

All values are given in kcal/mol, and results are intended to be interpreted comparatively, rather than as absolute values. When normalized according to dimer count, no significant differences are observed between the wild-type histone systems of varying sizes. The G17D mutation yields a slight reduction in average calculated binding strength, but this difference is not statistically significant (p-value of 0.157). Statistical significance was calculated via t-test, and quoted error values are the standard error of the mean calculated across the three independent trajectories.

| System | $\Delta G_{DNA}$ | $\Delta G_{DNA}$ (per dimer) |
|---|---|---|
| Arc90 | −459.4 ± 4.7 | −153.1 ± 1.6 |
| Arc120 | −610.5 ± 5.3 | −152.6 ± 1.3 |
| Arc120-G17D | −580.2 ± 15.5 | −145.0 ± 6.7 |
| Arc180 | −906.0 ± 5.6 | −151.0 ± 0.9 |

GA-MC analysis of the Arc207 traces shows the same correlation between $f/f_o$ and $MgCl_2$ concentration that is observed for the sedimentation rate, with apparent compaction from 1 to 5 mM $MgCl_2$ and no further reduction beyond 5 mM $MgCl_2$ (*Figure 5B*, top). A modest reduction in $f/f_o$ is observed for the Nuc147 sample, but the most notable differences between the archaeasome and nucleosome systems is the loss of nucleosome sample due to aggregation. For Nuc147, $OD_{260}$ values rapidly decline with increased $Mg^{2+}$, and no appreciable sample remains at 5 mM $MgCl_2$ and above (*Figure 5B*, bottom). In contrast, the Arc207 complex displays no significant losses, even at 10 mM $MgCl_2$. This shows that, unlike eukaryotic chromatin, divalent cations compact archaeasomes without promoting fiber-fiber association.

## Archaeasomes exist in several structural states

Single particle cryoEM was utilized to determine the three-dimensional structure of archaeasomes in the presence of 5 mM $MgCl_2$. Particles appeared more defined than without $MgCl_2$, and individual particles with tightly wrapped DNA conformations are distinguishable from other states where density is visible as perpendicular extensions to the wrapping plane (*Figure 6A*, blue and gold boxes). A total of 1,879,294 particles were identified according to a neural network trained on manual particle selections and then classified. The two-dimensional classification of this dataset identify two different populations of Arc207: Class I, which upon first inspection has the classical 'nucleosome-like' fold (*Figure 6B*), and Class II, in which two perpendicular DNA wrappings are clearly observed (*Figure 6C*). We separated these particles into independent 3D classes and refinement schemes, which yielded 80,609 particles in Class I and 5959 particles in Class II. Many more particles could have been included in each class, especially Class II, but we were conservative to ensure that no

**Table 2.** SV-AUC properties of Arc207 and Nuc147 molecules.

Despite the higher mass of the Arc207 complex, it sediments slower than the Nuc147 system due to increased particle elongation (frictional ratio, '$f/f_o$'). '$MW_{the}$' denotes the theoretical molecular weight, and '$MW_{GA-MC}$' corresponds to the molecular weight derived via GA-MC analysis. Parenthetical values outline the 95% confidence interval of each GA-MC value.

| | Nuc147 | Arc207 | 207 DNA |
|---|---|---|---|
| $MW_{the}$ (kDa) | 200.3 | 231.4 | 127.8 |
| $MW_{GA-MC}$ (kDa) | 218.6 (207.2, 229.9) | 266.5 (256.9, 276.1) | 156.6 (152.4, 160.1) |
| $f/f_o$ | 1.55 (1.49, 1.61) | 1.94 (1.89, 1.99) | 3.14 (3.08, 3.20) |
| Sedimentation Coefficient (S) | 11.2 (11.1, 11.2) | 10.6 (10.6, 10.7) | 6.1 (6.1, 6.1) |

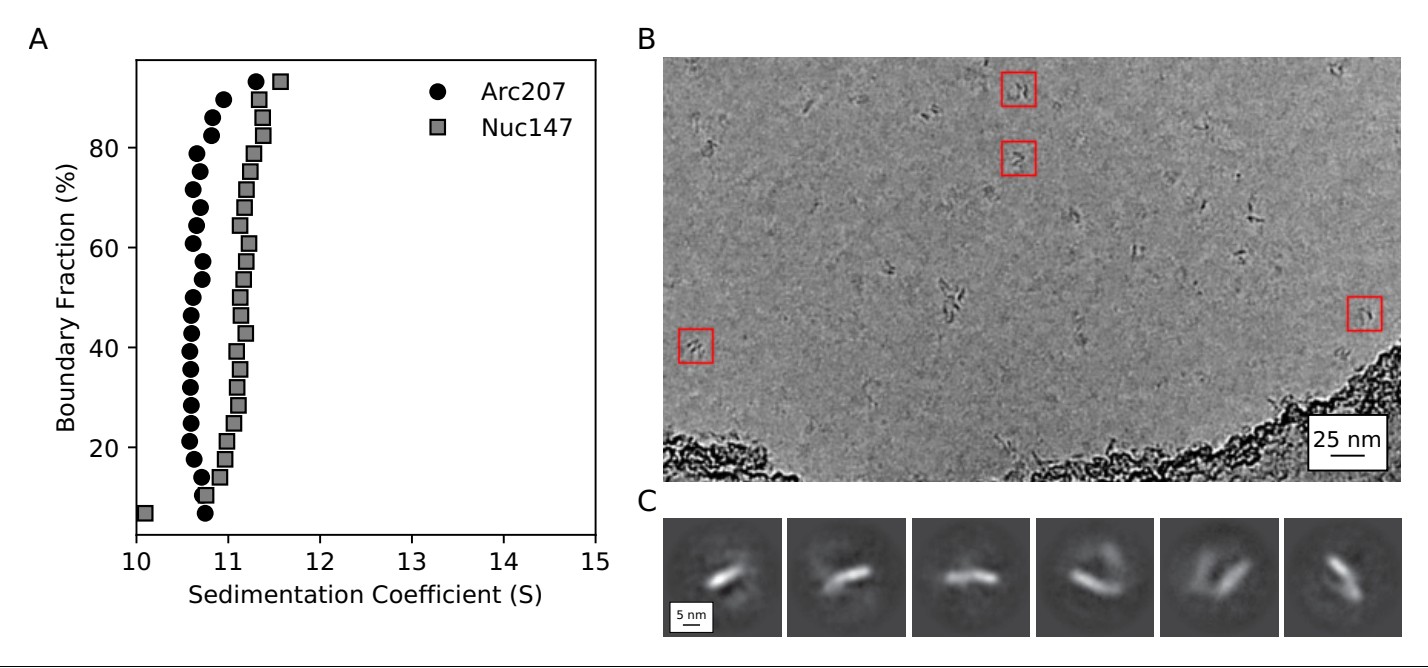

**Figure 4.** Biophysical characterization of archaeasomes in vitro. (**A**) van Holde-Weischet plots of Arc207 (circles) and Nuc147 (squares) samples. While Arc207 has a larger mass than the eukaryotic Nuc147, it sediments more slowly, indicative of increased drag caused by extended particle configurations. (**B**) Representative de-noised cryoEM micrograph of Arc207 sample. Particles are observed with DNA wrapped in a nucleosome-like pattern, but a significant separation between neighboring DNA turns is observed. (**C**) Two-dimensional classifications of particles extracted from these micrographs. No tightly wrapped DNA is identified, in agreement with our SV-AUC inferences of an extended Arc207 particle shape in comparison to compact Nuc147.

The online version of this article includes the following source data and figure supplement(s) for figure 4:

**Source data 1.** van Holde-Weischet data extracted from SV-AUC sedimentation profiles for *Figure 4A*.

**Figure supplement 1.** Titration of HTkA histone on to 207 bp DNA strand, observed by native PAGE gel.

interactions with neighboring particles affected their configuration. Refinement of these densities yielded maps at 9.5 Å and 11.5 Å resolution for Class I and Class II, respectively (*Figure 6D,E*).

At 9.5 Å resolution, secondary structure and side chain assignment is not possible for Class II, which has the higher resolution and larger particle count of the two classes. However, key features of the archaeasome previously seen in the crystal structure are clearly discernible from the density, such as DNA wrapping around a core of five histone dimers and periodic, overlapping densities that we associate with α2, the longest of the core α-helices in each histone (*Video 6*). Even though Arc207 (207 base pairs of DNA bound to seven histone dimers) were deposited on the grid, refined density of state Class II describes only 150 base pairs of DNA and five histone dimers (*Video 7*). As such, we trimmed the final frame from one of our MD simulations of the Arc180 complex down to an Arc150 complex and docked it to the map using the 'fit in map' function of Chimera. We simulated density at 8 Å resolution for the MD-derived structure, which yields a correlation coefficient of 0.85 with the empirical map. This shows that our MD simulations describe with high fidelity a compact archaeasome composed of five histone dimers wrapping ~150 bp of DNA.

The density for Class II has sufficient volume to be fit with the full 207 base pair DNA and seven histone dimers (*Video 8*). The larger portion of the volume is best described by four histone dimers binding ~120 base pairs of DNA in a nucleosome-like structure, and the smaller portion can be fit with three histone dimers binding ~90 base pairs. While the connecting density suggests only moderate bends in the helical axis, the wrapping pathways of the subunits result in the two faces being arranged at a near 90° angle. The continuity of the DNA between the two moieties was confirmed by modulating the electron density contours (*Video 9*). Structural models were generated by rigid-body docking separate Arc120 and Arc90 subunits into the associated density, and the connecting

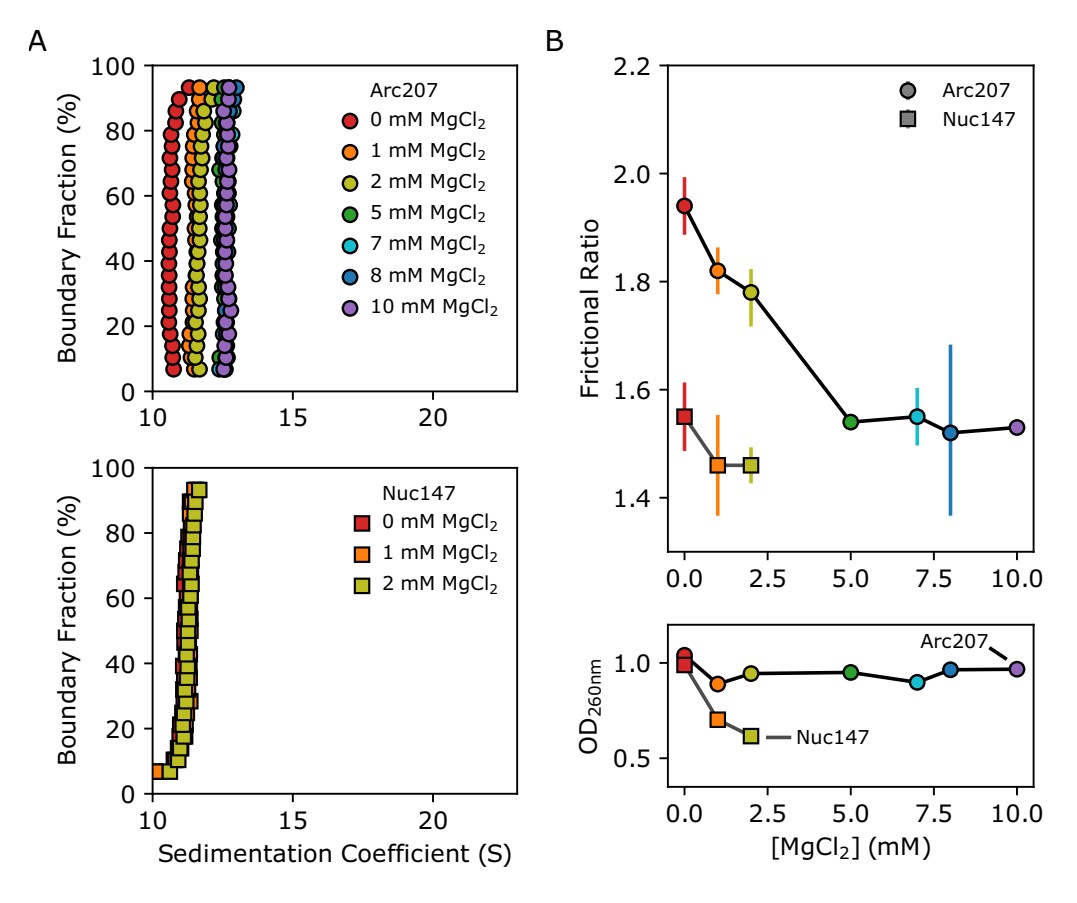

**Figure 5.** Effects of $Mg^{2+}$ concentration on sedimentation behavior of Arc207 and Nuc147 samples. (**A**) van Holde-Weischet plots of (top) Arc207 and (bottom) Nuc147 samples. Increases from 0 to 5 mM $MgCl_2$ results in an increase in Arc207 sedimentation rate, but little change in Nuc147 sedimentation. (**B**) Effects of $MgCl_2$ concentration on (top) frictional ratio and (bottom) absorbance at 260 nm for Arc207 (circles) and Nuc147 (squares). Frictional ratios of Arc207 samples follow the same profile as sedimentation rate, where samples compact from 0 to 5 mM $MgCl_2$ but with little change from 5 to 10 mM $MgCl_2$. Similarly, Nuc147 samples showed very modest changes in compaction. $OD_{260nm}$ measurements show Nuc147 particles aggregating as a result of increased $MgCl_2$ concentration, but Arc207 samples compact without losses to aggregation.

The online version of this article includes the following source data for figure 5:

**Source data 1.** van Holde-Weischet data extracted from SV-AUC sedimentation profiles as a function of $MgCl_2$ concentration for *Figure 5A*, as well as frictional coefficients (as determined by GA-MC analysis) and $OD_{260nm}$ values for *Figure 5B*.

DNA was energy minimized and subjected to a short MD simulation to confirm that the connecting DNA segment is reconcilable with B-form DNA parameters (*Figure 6—figure supplement 1*).

To confirm that the structures described by the EM densities also exist in solution, and to correlate these findings with our SV-AUC analyses of particle compaction, we utilized the SoMo plugin of UltraScan to calculate sedimentation properties from our derived models. For the fully resolved Arc207 model, with the Arc90 'lid' bending ~90° from the Arc120 'core', we calculate a frictional ratio of 1.53, in close agreement with the 1.55 value extracted from the experimental traces. For the particles that can be fit with five histone dimers and 150 bp DNA (Class I), we surmised that the remaining 60 base pairs of DNA (and two histone dimers) extend from the observed density but assume variable orientations with respect to the main particle resulting in very weak density. Close inspection of the two-dimensional classes in this category indeed shows additional out-of-plane density that is consistent with the missing two dimers and ~60 base pairs of DNA, albeit at low contrast relative to background (*Figure 6B*). Further three-dimensional classifications of the 9.5 Å density reduced the overall resolution but yielded indications of the missing volume, albeit at low fidelity (*Figure 6—figure supplement 2* and *Video 10*). To further model this particle, we extended the Arc150 structure with an additional Arc60 subunit positioned 90° out-of-plane, similar to the model

derived for the fully observed Arc207 density. This model predicted a similar value for the frictional coefficient of 1.50. In contrast, models with continuous wrapping and no deflections in the DNA pathway (as observed in the crystal structure, but not populated significantly in the cryo EM images) yield a frictional coefficient of 1.36. These data together show that archaeasomes on an extended DNA fragment can be viewed as a distribution of archaeasome subunits that can open to a ~90° angle.

## Discussion

Using in silico and in vitro approaches, we investigated the structure and dynamics of the 'slinky-like' architecture of histone-based archaeal chromatin. SV-AUC and cryoEM both show that the archaeasome, containing more than the characteristic four histone fold dimers observed in nucleosomes, is inherently dynamic and exists in a variety of open states while maintaining DNA-histone interactions. Archaeasomes can be stabilized with divalent cations. In apparent absence of archaeal ATP-dependent chromatin remodeling factors (large machines that regulate chromatin access in eukaryotes), this architecture provides an alternative mechanism for compacting chromatin and maintaining genome accessibility.

Our cryoEM structures show that archaeasomes are metastable and extended in the absence of divalent cations. Even in the presence of 5 mM $Mg^{2+}$, archaeasomes exhibit at least two configurations. Indeed, our structures contain a maximum of four to five histone dimers arranged in a continuous helical ramp. Beyond that, the helical histone ramp is disrupted at the four-helix bundle interface by an outward rotation of the remaining two to three histone dimers and attached DNA. The existence of these structures in solution is supported by analysis of the hydrodynamic parameters of archaeasomes obtained from SV-AUC. Extending this arrangement to even longer DNA, one could picture particles that wrap anywhere from ~90 to~150 base pairs (using three to five histone fold dimers) arranged at ~90 degree angles with respect to each other, connected by minimal linker DNA. This arrangement may leave the major groove of the connecting DNA susceptible to nuclear factors, such as micrococcal nuclease or transcription factors, and explains the ~30 bp MNase digestion ladder observed in native archaeal chromatin.

Consistent with this interpretation, molecular dynamics simulations of Arc90 assemblies (with three histone fold dimers) displayed dynamic breathing at terminal sites of the complex, while histone-histone and histone-DNA contacts are maintained. Models constructed from repeats of this conformation agree with particle conformations extracted from SV-AUC hydrodynamic parameters, as well as direct images and two-dimensional class averages from cryoEM data gathered in the absence of $Mg^{2+}$. There, multiple turns of DNA, fully saturated with histones, were observed but with considerable flexibility and distance between neighboring gyres. In contrast, Arc120 and Arc180 simulations (containing four and five histone dimers, respectively) did not exhibit this degree of dynamic behavior. However, these simulations may have been artificially biased for the closed configuration, because the starting structures had intact L1-L1 interactions. Separation of these contacts may require more than several hundred nanoseconds of computational sampling, and simulations of the Arc120-G17D system indeed show appreciable separation between DNA turns by weakening this interface while maintaining four-helix bundle interactions.

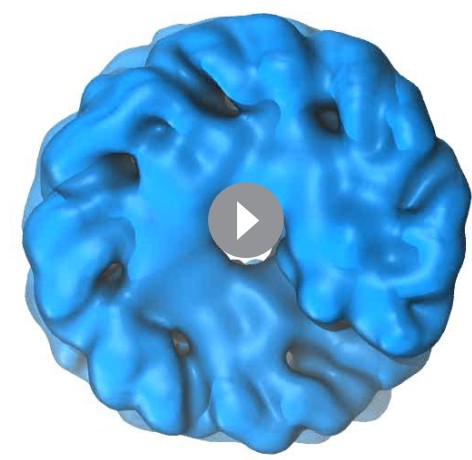

**Video 6.** Derived EM density for the Class I state of the Arc207 complex. By varying the contour levels of the density, we find that the strongest contributors to the EM volume are around the outside of the molecule (attributed to DNA), as well as overlapping periodic densities in the histone core (attributed to each histone's α2 helix).

https://elifesciences.org/articles/65587#video6

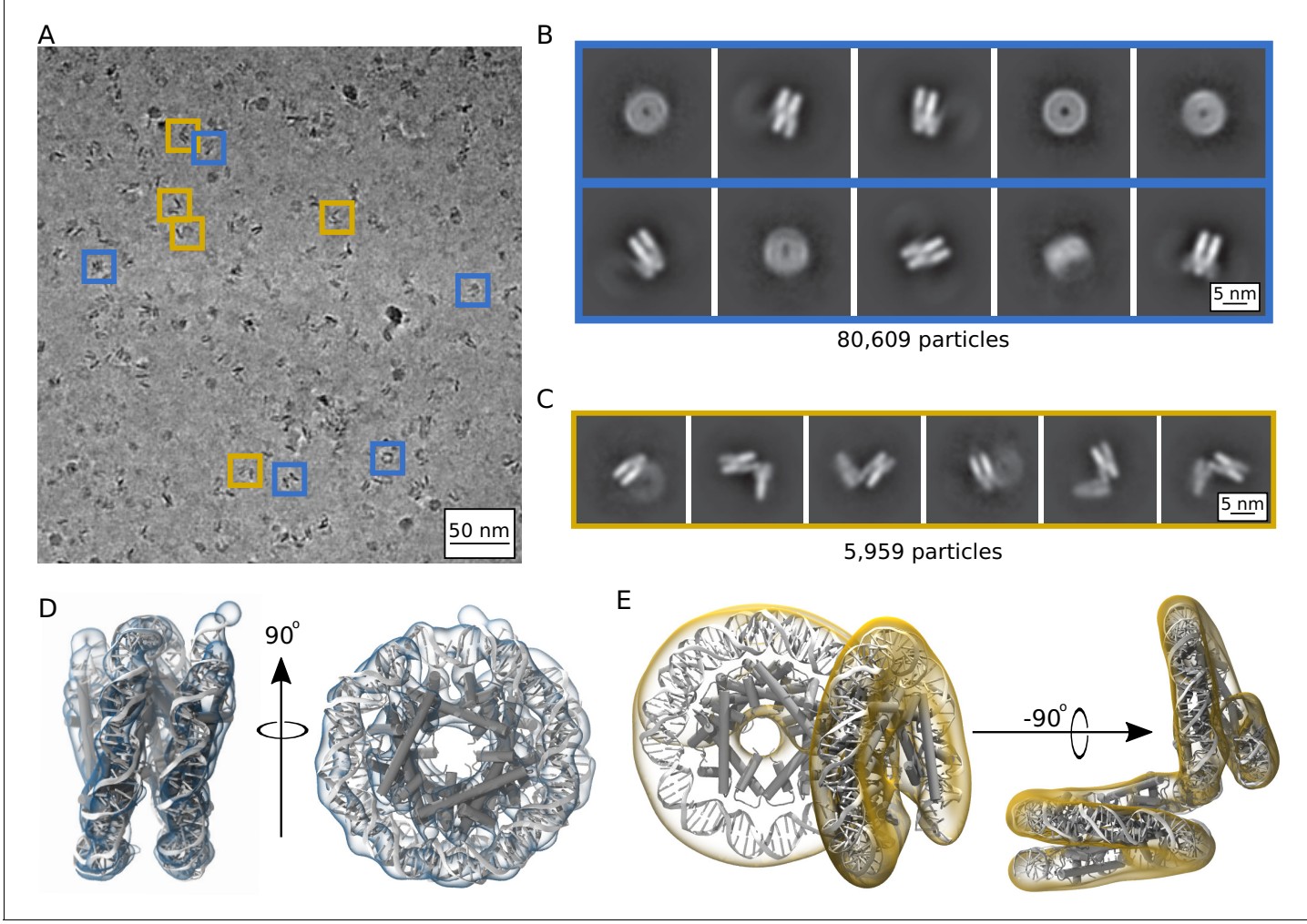

**Figure 6.** Single particle cryoEM analysis of the Arc207 complex in the presence of 5 mM MgCl$_2$. (**A**) Representative denoised micrograph. Particles are observed with both 'nucleosome-like' wrappings of the 207 base pair DNA fragment (blue boxes, Class I) as well as particles showing out-of-plane 'lid' extensions of DNA from similarly wrapped 'cores' (gold boxes, Class II). Also shown are two-dimensional classes (**B**, Class I) and (**C**, Class II). Three-dimensional densities from these classes are shown in (**D**) and (**E**), respectively. Class I particles were fit by rigid-body docking of Arc150 coordinates extracted from Arc180 simulations, and Class II particles were fit by rigid-body docking separate Arc90 and Arc120 components and bridging the connecting DNA through energy minimization.

The online version of this article includes the following figure supplement(s) for figure 6:

**Figure supplement 1.** DNA parameters of the zero-length linker segment calculated from a 10 ns implicit solvent simulation of the Arc207 system restrained to the 'Class II' configuration.

**Figure supplement 2.** Low fidelity density of Arc150 volume subclassification, showing a potential Arc60 'lid' oriented out-of-plane with Arc150 'base' reconstructed from the 'Class I' particles.

In eukaryotes, chromatin structure can be further modulated by the incorporation of histone variants (*Bönisch and Hake, 2012*). While in vivo studies have identified disparate roles of variant histones in archaea (*Čuboňováa et al., 2012*), very little is known about the structural modifications that they invoke. In a recent study by *Stevens et al., 2020*, molecular dynamics trajectories showed that substituting major type histones in *Methanosphaera stadtmanae* with a 'capstone' paralog, a histone variant with sequence modifications at the four-helix bundle, disrupts the tetramer interface and separates histone dimers on a continuous DNA fragment, thereby limiting archaeasome size. In the context of the capstone model, our EM-derived structures suggest that destabilizing the four-helix bundle interface may yield a subtle increase in DNA accessibility through increased exposure of the major groove, and without the need to fully displace the archaeasome subunit. As we have shown,

this dynamic behavior is already sampled by major-type archaeal histones deposited on a continuous DNA fragment, and substitution of terminal histones with capstone-containing dimers will further bias the assembly toward the open configuration. In this way, lengths of DNA that would typically contain a single archaeasome unit of considerable length would instead be segmented in to several sub-archaeasomes configured in repeat perpendicular arrangements that modestly expose bridging DNA segments. Additionally, high rigidity in certain DNA sequences may inherently poise these segments as linker sequences, and the frequency of these less flexible regions would similarly influence the number of 90°-oriented subunits present along the genome.

Our SV-AUC traces show that Arc207 assemblies exhibit no signs of self-association upon the addition of up to 10 mM $MgCl_2$, unlike eukaryotic nucleosomes and nucleosome arrays, where this favors extensive inter-nucleosome and inter-array interactions and aggregation (*Schwarz et al., 1996*). Instead, these conditions bias individual archaeasomes toward more compact states without promoting inter-archaeasome interactions. Interestingly, estimates of $Mg^{2+}$ concentration within *T. kodakarensis* cells have been quoted at ~120 mM (*Nagata et al., 2017*), much higher than measurements of $Mg^{2+}$ in mammalian nuclei (16–18 mM) (*Romani, 2013*). Per dimer, archaeal histones provide weaker electrostatic screening of DNA than eukaryotic histones, due to their relatively lower isoelectric points (pI of ~8 vs ~11), and the greater overall negative charge of archaeasomes may encourage rapid sequestering of any free $Mg^{2+}$ ions in the cell. This would induce a gradient-based mechanism for rapid and excessive uptake of cations to prevent chromatin compaction from monopolizing the $Mg^{2+}$ necessary for enzyme action. On the other hand, the ability of *T. kodakarensis* archaeasomes to compact but not aggregate as a result of elevated $Mg^{2+}$ concentration may be a direct result of evolving in high-salt environments, as regulating between internal and external concentrations may have been more energetically demanding than simply adapting their chromatin structure. More specific measures of local $Mg^{2+}$ concentration in archaeal chromatin may help differentiate these two mechanisms.

Archaeal histones were successfully expressed in *E. coli* cells and were found to coat bacterial DNA (*Rojec et al., 2019*). Surprisingly, this resulted in only minor perturbations of cell growth, despite *E. coli* lacking the evolutionary machinery to handle histones. Furthermore, in vitro assays have shown that transcription through archaeasomes is slowed but not

**Video 7.** EM density for the Class II state of the Arc207 complex (clear blue) docked with Arc150 complex (gray) extracted from MD simulations of Arc180 complex. Docking of the Arc150 construct shows that this density cannot capture the full complex deposited on the grid but only ~150 base pairs of DNA and five histone dimers. Additional density can be seen to extend along the DNA path beyond the bound core, suggesting a possible continuation of linker DNA that could extend to the potential tetramers observed in the 2D classes (*Video 10*, see main text for details).
https://elifesciences.org/articles/65587#video7

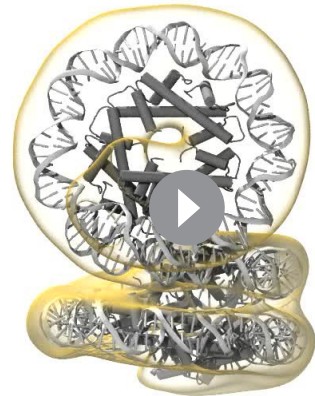

**Video 8.** EM density of the Class II state of the Arc207 complex (clear gold) docked with two complexes (~Arc120 and ~Arc90) connected by short linker DNA (gray). The ~Arc90 complex is bent ~90° out of plane of the core Arc117 complex.
https://elifesciences.org/articles/65587#video8

altogether stopped (*Xie and Reeve, 2004*). In eukaryotes, navigation of polymerases through chromatin is assisted by histone chaperones and ATP-dependent remodeling factors (*Hammond et al., 2017*; *Clapier et al., 2017*; *Markert and Luger, 2021*), but no known homologs to these complexes have been identified as yet in archaea. In absence of remodelers, the inherent dynamics of archaeasome-based chromatin may thereby allow limited access to chromatin by the sporadic (and possibly stochastic) appearance of near-zero linker DNA and Arc60 or Arc90 substates. On the other hand, it is possible that sequences encoding chromatin remodelers are yet to be found. On a related note, the 'SMC-like' coalescin proteins was recently identified in the histone-less *Sulfolobus* archaea (*Takemata and Bell, 2021*), and the access of these yet unidentified 'hidden agents' would similarly be tuned by the dynamics of the archaeasome. The degree to which DNA rigidity, variant histones, or unknown histone- and DNA-binding proteins regulate chromatin accessibility is an exciting future area of research'.

## Materials and methods

### Molecular dynamics simulations

Archaeasome systems of varying sizes were constructed from the crystal structure containing ~90 bp of DNA bound to histone HMfB (PDB 5T5K). Systems containing ~90 base pairs of DNA and three histone dimers (Arc90), ~120 base pairs of DNA and four histone dimers (Arc120), and ~180 base pairs of DNA and six histone dimers (Arc180) were created as prescribed by the crystal lattice. The Arc90 system is intended to represent the 'fundamental unit' of archaeasome-based chromatin, as it is the crystallographic unit of the solved structure as well as the smallest DNA protection footprint observed by MNase digestion (*Mattiroli et al., 2017*). The Arc120 and Arc180 complexes provide systems in which to study the contribution of stacking histone-histone interactions, as well as the wrapping of additional DNA superhelical turns, in stabilizing or destabilizing the proposed archaeasome. Additionally, an Arc120 system with the G17D mutation was also generated (Arc120-G17D), in analogy to the destabilizing G17D mutation that was previously studied in vivo. The overhanging DNA bases that formed crystallographic contacts were removed in our simulations. Each system was neutralized and solvated in a TIP3P box of 100 mM NaCl (*Jorgensen et al., 1983*), and masses were repartitioned from heavy atoms to covalently bonded hydrogen atoms to allow for the use of a four fs timestep (*Hopkins et al., 2015*). Parameters for protein atoms were taken from the Amber FF14SB forcefield (*Maier et al., 2015*), DNA parameters were taken from the Amber bsc1 forcefield (*Ivani et al., 2016*), and ions were parameterized according to the modifications of *Joung and Cheatham, 2008*.

Systems were then energy minimized for 5000 steps while constraining solute heavy atoms with a 10 kcal/mol/Å² harmonic potential, followed by 5000 steps without restraints. After energy minimization, three independent simulations of each system were conducted according to the following protocol: simulations were heated from 10 K to 300 K over the course of 50 ps in the NVT ensemble with heavy atom restraints applied, system densities were equilibrated and positional restraints were slowly released over the course of 200 ps in the NPT ensemble (target pressure of 1 atm), and simulations were extended for 300 ns without positional restraints in the NPT ensemble. During these production simulations,

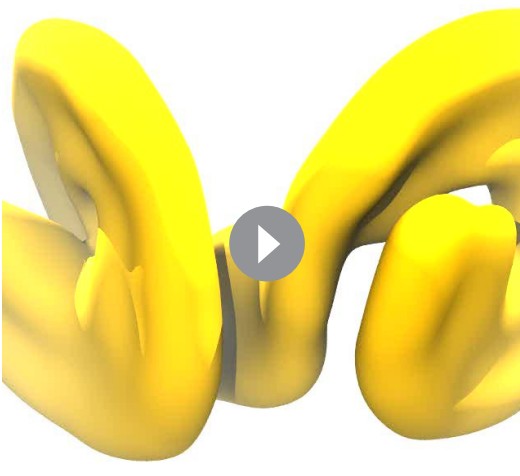

**Video 9.** EM density flaring of the Class II state, showing that the linker DNA portion is continuous along the length of the Arc207 complex.
https://elifesciences.org/articles/65587#video9

terminal base pair fraying was removed from the simulation by reinforcing hydrogen bonding in the terminal base pairs. No restraints were applied when participating atoms were within 3.5 Å of one another, but a harmonic potential was applied when participating atoms spread beyond this cutoff (force constant = 5.0 kcal/mol/Å$^2$). Minimization and MD simulations were conducted in the pmemd engine (v18), with CUDA acceleration utilized for the simulations (*Salomon-Ferrer et al., 2013*).

## Simulation analysis

System equilibration was monitored through RMSD calculations of backbone atom positions, and local flexibility was measured via the root mean-squared fluctuation (RMSF) of the backbone. DNA breathing dynamics were quantified using the center of mass distance between the terminal base pairs and the neighboring superhelical turn. Complex stabilities were assessed using the MM-GBSA method with the igb5 solvent parameters and mbondi2 atomic radii (*Onufriev et al., 2004*). Trajectories and single frames were rendered using VMD, and structural analyses (RMSF, RMSD, DNA breathing) were calculated using cpptraj. Statistical significance between populations were determined by unpaired t-test.

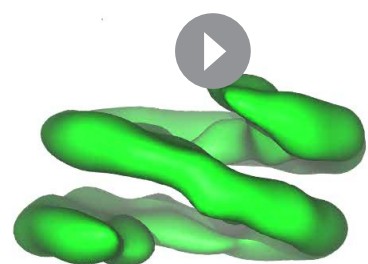

**Video 10.** EM density flaring of the Arc150 density subclassification of Class I particles, identifying weak Arc60-like density that can be associated out-of-plane orientations.
https://elifesciences.org/articles/65587#video10

## In vitro sample preparation

HTkA histones were provided by the Histone Source (CSU Fort Collins, CO), and the 207 base pair, Widom 601-derived sequence was purified as described (*Mattiroli et al., 2017*; *Dyer et al., 2003*). Archaeasome complexes (Arc207) were formed by mixing 207 bp DNA with HTkA histones at a 1:7 molar ratio of DNA to histone dimer in a buffer containing 100 mM KCl and 50 mM Tris (pH 8.0) and incubated for 20 min at room temperature. Samples were then dialyzed in buffers containing 0, 2, 5, 7, or 10 mM MgCl$_2$ at a volume ratio of 1:1000 two times, first for 2 hr and then overnight, as preparation for subsequent measurements. As histone stocks are suspended in high glycerol, the dialysis process simultaneously served to effectively remove glycerol.

## SV-AUC

Sedimentation velocity analytical ultracentifugation (SV-AUC) measurements were conducted in absorbance mode ($\lambda$ = 260 nm). Samples were loaded in to an An60Ti rotor in 400 µL cells with two-channel Epon centerpieces and then spun at 35,000 rpm at 20°C in a Beckman XL-A ultracentrifuge. Partial specific volumes of the samples were determined using UltraScan3 (v4.0) (*Demeler, 2005*). Time and radially invariant noises were subtracted through two-dimensional sediment analysis (2DSA), and the final 2DSA model parameters were used to initialize genetic algorithm and Monte Carlo analyses. Sedimentation coefficients were determined using van Holde-Weischet analysis (in Svedberg units, corrected to solvent conditions of water at 20°C), and molecular weights and frictional ratios (f/f$_o$ - the degree of elongation/flexibility in comparison to ideal spherical particles) were determined from genetic algorithm analysis of SV-AUC traces. Theoretical f/f$_o$ values for modeled structures were derived using the SoMo plugin of UltraScan, where bead models were created through the 'SoMo Overlap' scheme and hydrodynamics were calculated with the ZENO algorithm (*Brookes et al., 2010*; *Douglas et al., 1994*).

## CryoEM grid preparation and collection

After overnight dialysis (100 mM KCl, 50 mM Tris pH 8, 5 mM MgCl$_2$), Arc207 samples were concentrated to 0.9 mg of DNA per mL, and 4 μL of sample was deposited onto glow-discharged grids (40 mA for 45 s). Samples were screened on copper Cflat (1.2/1.3) grids, and formaldehyde-cleaned Quantifoil (2/2) grids were used for final data collection in order to increase image acquisition speed through a '2 × 2' collection scheme, where a single defocus level is used for a cluster of four grid holes. Each grid was manually plunge frozen in liquid ethane and stored in liquid nitrogen. Datasets were collected on a FEI Tecnai F20 equipped with a Gatan K3 camera at 29,000x magnification in counting mode (yielding 1.291 Å per pixel), 200 kV accelerating voltage, dosage rate of ~1 e$^-$/Å$^2$ per frame, and 50 frames per micrograph stack (total dose of 50 e$^-$/Å$^2$). A total of 5388 image stacks were collected for subsequent three-dimensional analysis. An Elsa Cryo-Transfer Holder (Gatan, Inc) and defocus range of −1.2 to −2.6 μm was used.

## Single particle analysis of CryoEM data

Using the Relion interface (v3.0) (*Zivanov et al., 2018*), micrographs were motion corrected using MotionCorr2 (*Zheng et al., 2017*), and CTF parameters were estimated from gCTF (*Zhang, 2016*). Ten micrographs were then randomly selected from a collection of motion-corrected micrographs and denoised using the janni_denoise.py function of the SPHIRE-crYOLO particle-picking pipeline (*Wagner, 2019*). Particles were manually picked from these denoised micrographs and used to train a neural network through crYOLO for picking particle coordinates across all micrographs, where the final particle predictions utilized the same denoising process (*Wagner et al., 2019*).

Densities shown here are the result of refinements to the largest dataset, collected in the 2 × 2 scheme. In total, 1,879,294 particles were predicted by crYOLO and extracted from the motion-corrected (non-denoised) micrographs using Relion and imported to CryoSparc (v2.12.4) for two-dimensional (2D) classification (*Punjani et al., 2017*). 2D classes were manually filtered to remove particles containing primarily noise or interactions with overlapping neighbor particles. Subsequent 2D classifications showed two predominant particle types: the 'Class I' archaeasome state characterized by tightly wound DNA, and the 'Class II' archaeasome state showing nucleosome-like DNA arrangements with distinct out-of-plane densities, forming 'base' and 'lid' subunits. These classes were then separated from one another for three-dimensional (3D) classification and refinement by CryoSparc. Class I density was refined through Bayesian Polishing of the contributing particles in Relion, but the Class II density saw no benefit from polishing and the CryoSparc-derived density is reported. Reconstructed volumes were deposited to the Electron Microscopy Data Bank (EMD-23403, EMD-23404). Figures and videos were generated with VMD (v1.9.3) (*Humphrey et al., 1996*).

## Modeling of EM densities

Class I density (containing volume of five histone dimers and ~150 bp of DNA) was modeled by extracting five dimers and ~150 bp of DNA from the end state of a randomly selected Arc180 simulation. Simulation coordinates were fit to the EM-derived density through rigid body docking in Chimera. The Class II density was modeled by first docking Arc90 and Arc117 constructs in the smaller and larger volumes, respectively. Then, the bridging DNA segments were ligated via tleap and energy-minimized via pmemd in implicit solvent (Amber FF14SB protein forcefield, DNA bsc1 parameters, and igb5 implicit solvent model with mbondi2 atomic radii modifications and a 100 mM monovalent salt environment). Calculation of nucleic acid geometry parameters was conducted with cpptraj.

## Acknowledgements

We thank Garrett Edwards for valuable conversations regarding SV-AUC experiments and analyses, as well as Pamela Dyer and Uma Muthurajan for discussions about sample preparation. We also thank Alison White for purified DNA stocks, and Kathryn M Stevens and Tobias Warnecke for constructive feedback on this work.

Work in the Weresczynski group was supported by an NSF Career Award (1552743) and an NIH NIGMS R35 award (R35GM119647), and funding for the Luger lab is provided by the Howard Hughes Medical Institute (HHMI). SB was funded in part by both HHMI and the NIGMS R35 Award

to JW, as well as by an NIH NIGMS NRSA fellowship (F32GM137496). The contents of this article are the sole responsibility of the authors and do not necessarily represent the official views of the National Institutes of Health.

MD simulations were conducted, in part, on an SDSC Comet GPU allocation to SB through the XSEDE environment, which is supported by NSF grant ACI-1548562 (*Towns et al., 2014*). Electron microscopy was done at the University of Colorado, Boulder, EM Services Core Facility in the MCDB Department, with the technical assistance of facility staff. Calculations for modeling SV-AUC data were performed on the UltraScan LIMS cluster at the Bioinformatics Core Facility at the University of Texas Health Science Center at San Antonio and multiple high-performance computing clusters supported by NSF XSEDE Grant #MCB070038 (to Borries Demeler).

## Additional information

### Funding

| Funder | Grant reference number | Author |
|---|---|---|
| National Science Foundation | 1552743 | Jeff Wereszczynski |
| National Institute of General Medical Sciences | R35GM119647 | Jeff Wereszczynski |
| Howard Hughes Medical Institute | | Karolin Luger |
| National Institute of General Medical Sciences | F32GM137496 | Samuel Bowerman |

The funders had no role in study design, data collection and interpretation, or the decision to submit the work for publication.

### Author contributions

Samuel Bowerman, Conceptualization, Resources, Data curation, Formal analysis, Funding acquisition, Validation, Investigation, Visualization, Methodology, Writing - original draft, Writing - review and editing; Jeff Wereszczynski, Conceptualization, Resources, Supervision, Funding acquisition, Writing - original draft, Writing - review and editing; Karolin Luger, Conceptualization, Resources, Supervision, Funding acquisition, Validation, Writing - original draft, Project administration, Writing - review and editing

### Author ORCIDs

Samuel Bowerman (iD) https://orcid.org/0000-0003-0753-4294
Jeff Wereszczynski (iD) https://orcid.org/0000-0002-2218-3827
Karolin Luger (iD) https://orcid.org/0000-0001-5136-5331

### Decision letter and Author response

Decision letter https://doi.org/10.7554/eLife.65587.sa1
Author response https://doi.org/10.7554/eLife.65587.sa2

## Additional files

### Supplementary files

- Supplementary file 1. Structural model for Class I particles (pdb format).
- Supplementary file 2. Structural model for Class II particles (pdb format).
- Transparent reporting form

## Data availability

cryoEM datasets have been uploaded to EMPIAR (EMD-23403, EMD-23404). The pdb files are submitted as supplementary information. MD trajectories will be stored on CU storage resources (PetaLibrary) and made available upon request through file transfer or shipping of external hard drives.

The following datasets were generated:

| Author(s) | Year | Dataset title | Dataset URL | Database and Identifier |
|---|---|---|---|---|
| Bowerman S, Wereszczynski J, Luger K | 2021 | Reconstituted 207 bp Archaeasome, Class I | http://www.ebi.ac.uk/pdbe/entry/emdb/EMD-23403 | Electron Microscopy Data Bank, EMD-23403 |
| Bowerman S, Wereszczynski J, Luger K | 2021 | Reconsituted 207 bp Archaeasome, Class II | http://www.ebi.ac.uk/pdbe/entry/emdb/EMD-23404 | Electron Microscopy Data Bank, EMD-23404 |

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
