## [Decision Letter]

**Acceptance summary:**

The authors present an elegant combination of cryo-EM, analytical ultracentrifugation and molecular dynamics simulations to investigate the structure and dynamics of archaeal histone – DNA complexes, termed archaeasomes to distinguish them from eukaryotic nucleosomes. This rigorous biophysical study provides important new insights into archaeal genome biology and will no doubt be of interest to the archaeal research community as well as the chromatin biology field in general.

**Decision letter after peer review:**

Thank you for submitting your article "Archaeal chromatin 'slinkies' are inherently dynamic complexes with deflected DNA wrapping pathways" for consideration by *eLife*. Your article has been reviewed by two peer reviewers, including Sebastian Deindl as the Reviewing Editor and Reviewer #1, and the evaluation has been overseen by Cynthia Wolberger as the Senior Editor. The following individual involved in review of your submission has agreed to reveal their identity: Yamini Dalal (Reviewer #2).

Essential Revisions:

This is a very strong manuscript. Please consider the reviewers' recommendations and suggestions below by making the appropriate textual changes. No additional experiments are required.

Reviewer #1 (Recommendations for the authors):

It is not completely clear why the authors chose histones from *Thermococcuskodakarensis* for the experimental part of the work, while the previously published crystal structure and the simulations presented in this study used histones from *Methanothermus fervidus*. Can the authors comment more on this choice? Are the histones from one species are easier to express and purify than those from the other species? The text would benefit from a brief explanation, although this is not a concern for the validity of the study, given the high degree of sequence identity between these histones.

"consistent with a systemic error"

Do the authors mean "systematic" (as opposed to random) error?

In Table 2, is there a 95% confidence interval to report for sedimentation coefficients? If not, can the authors explain why?

"A total of 1,879,294 particles were identified and classified according to a neural network trained on manual particle selections"

This statement is a bit unclear. Is it particle picking that was performed using a neural network (crYOLO, as the Materials and methods section indicates), or is it the selection of good 2D class averages (which is also possible, using the program Cindirella from the same authors who make crYOLO)? If the former, I suggest mentioning the neural network before the classification procedure; this sentence as written now could be understood as if a neural network was used to select good 2D class averages (which is not what the Materials and methods section explains).

"Refinement of these densities yielded maps at 9.5 Å and 11.5 Å resolution for the closed and open forms, respectively (Figure 6E, F)."

There is no panel F in Figure 6. Earlier references to Figure 6 in the same paragraph also seem to be off by one panel.

"Even though Arc207 (207 base pairs of DNA bound to 7 histone dimers) were deposited on the grid, our refined density of the closed state describes only 150 base pairs of DNA and 5 histone dimers"

Is it possible that what the authors call the closed conformation is in fact the "base" of an open conformation, and that this reconstruction was obtained from truly open particles in which the "lid" was more dynamic and therefore not resolved? This would explain the missing molecular weight in this reconstruction. This possibility does not change the main conclusion that archaeasomes are dynamic, but it could change the way the 90 degrees open state is seen: it could correspond to an intermediate between the fully open and fully closed conformations, with the "lid" resolved because it is more stable than in the majority of the particles, that are fully open states in which the "lid" cannot be resolved, and the dataset may not contain many (any?) truly closed particles (which one would expect to yield a reconstruction showing all 207 base pairs of DNA and 7 copies of the histone dimer). The authors should comment on this.

"Close inspection of the "closed state" two-dimensional classes indeed shows additional out-of-plane density that is consistent with the missing two dimers and ~60 base pairs of DNA, albeit at low contrast relative to background (Figure 7C)"

There is no Figure 7. This statement likely refers to Figure 6C.

"Archaeasome compaction can be stabilized with divalent cations. In absence of archaeal ATP-dependent chromatin remodeling factors (large machines that regulate chromatin access in eukaryotes), this architecture provides an alternative mechanism for compacting chromatin and adjusting genome accessibility"

There are several examples of regulatory mechanisms for chromatin remodeling in eukaryotes that involve domains which bind to specific histone post-translational modifications as a way to target remodeling activity, making it a mostly deterministic process (presence of a PTM causes recruitment of a remodeler). It is difficult to envision that any regulation mechanism in archaea could emerge from random conformational changes in their chromatin only. Is it definitely established from whole-genome sequencing that archaea do not have chromatin remodelers? Or could they have them, but these proteins have yet to be identified? Could the compaction induced by divalent cations be the main regulatory mechanism in vivo? (in which case membrane ion channels would also indirectly act as chromatin remodelers on a genome-wide scale by regulating intracellular concentrations of these ions?). The authors should consider discussing these points to enrich the Discussion section and potentially strengthen the connection between the their biophysical results and archaeal genome biology.

Materials and methods section, cryoEM grid preparation

The glow discharge conditions should be indicated.

"a Gatan K3 camera at 29,000x magnification in non-super resolution mode"

This mode can be called "counting mode".

"dosage rate of ~1 e/Å, and 50 frames per micrograph stack"

The dose rate's unit is e/Å2/s, unless the authors are referring to the dose per frame (in which case it should be clarified). In addition, the total dose accumulated over the entire exposure time should be indicated.

When first mentioning the Widom 601 sequence, the authors should cite (Lowary and Widom, 1998).

"Gel shift assays showed that full complex saturation occurs when DNA and histones are mixed at the previously reported stoichiometric limit"

This statement should cite an adequate reference.

Reviewer #2 (Recommendations for the authors):

It would be nice if they could include the rationale to study dimer-dimer and dimer-DNA interactions by MD using 90 bp, 120 bp and 180 bp.

A MNase ladder is mentioned, but no citation or figure is referenced, please add the MNase ladder.

The authors very nicely show that Mg^2+^ does not impact archaeasome oligomerization, how does salt concentration impact oligomerization?

The authors mention that particles that interacted with neighboring particles were not used in the analysis. We are curious whether these particles would more closely resemble what would happen in cells where this is higher density (one presumes) of chromatin?

What was the distribution of the angle of the open conformation found for the archaeasome?

It would be interesting and timely if the authors could discuss their results in light of the recent publication of the archaea genome organization by Takemata and Bell, 2020.

---

## [Author Response]

Essential Revisions:This is a very strong manuscript. Please consider the reviewers' recommendations and suggestions below by making the appropriate textual changes. No additional experiments are required.Reviewer #1 (Recommendations for the authors):It is not completely clear why the authors chose histones from Thermococcus kodakarensis for the experimental part of the work, while the previously published crystal structure and the simulations presented in this study used histones from Methanothermus fervidus. Can the authors comment more on this choice? Are the histones from one species are easier to express and purify than those from the other species? The text would benefit from a brief explanation, although this is not a concern for the validity of the study, given the high degree of sequence identity between these histones.

First, *T. kodakarensis* is genetically accessible while *M. fervidus* is not, and this prompted us to switch the system in our original paper (Mattiroli et al., 2017). Furthermore, *T. kodakarensis* is a well-established model organism for archaea (Rashid and Aslam, 2020). John Reeve, our original collaborator on this project who provided us with the HMfB protein preparations, has since retired, and *T. kodakarensis* histones were easier to prepare. The simulations were conducted during the transition between S.B.’s Ph.D. training in the Wereszczynski Lab and his initial training in benchtop biochemistry in the Luger Lab, so the computational models were constructed from the available data at that time – namely the HMfB crystal structure. We believe that laying out all this detail would be a distraction since the two histones have high sequence homology.

"consistent with a systemic error"Do the authors mean "systematic" (as opposed to random) error?

The reviewer’s interpretation is correct, and we have made this change in the manuscript.

In Table 2, is there a 95% confidence interval to report for sedimentation coefficients? If not, can the authors explain why?

Omitting the 95% confidence intervals in sedimentation coefficient was indeed an oversight on our part, and we have now included them in the table. Notably, they are very tight (~0.1 S) and thus our statements or conclusions remain fully supported by the data.

"A total of 1,879,294 particles were identified and classified according to a neural network trained on manual particle selections"This statement is a bit unclear. Is it particle picking that was performed using a neural network (crYOLO, as the Materials and methods section indicates), or is it the selection of good 2D class averages (which is also possible, using the program Cindirella from the same authors who make crYOLO)? If the former, I suggest mentioning the neural network before the classification procedure; this sentence as written now could be understood as if a neural network was used to select good 2D class averages (which is not what the Materials and methods section explains).

We thank the reviewer for pointing out some confusing language. Indeed, crYOLO was used to pick the 2D particles, but 2D classification was conducted with cryoSPARC, and not with the Cindirella program (although, we thank the reviewer for bringing this program to our attention). We have clarified the text:

“A total of 1,879,294 particles were identified according to a neural network trained on manual particle selections and then classified.”

"Refinement of these densities yielded maps at 9.5 Å and 11.5 Å resolution for the closed and open forms, respectively (Figure 6E, F)."There is no panel F in Figure 6. Earlier references to Figure 6 in the same paragraph also seem to be off by one panel.

Apologies. These were remnants from a previous draft of the manuscript that had escaped our proof reading. All figure references have been carefully confirmed and corrected.

"Even though Arc207 (207 base pairs of DNA bound to 7 histone dimers) were deposited on the grid, our refined density of the closed state describes only 150 base pairs of DNA and 5 histone dimers"Is it possible that what the authors call the closed conformation is in fact the "base" of an open conformation, and that this reconstruction was obtained from truly open particles in which the "lid" was more dynamic and therefore not resolved? This would explain the missing molecular weight in this reconstruction. This possibility does not change the main conclusion that archaeasomes are dynamic, but it could change the way the 90 degrees open state is seen: it could correspond to an intermediate between the fully open and fully closed conformations, with the "lid" resolved because it is more stable than in the majority of the particles, that are fully open states in which the "lid" cannot be resolved, and the dataset may not contain many (any?) truly closed particles (which one would expect to yield a reconstruction showing all 207 base pairs of DNA and 7 copies of the histone dimer). The authors should comment on this.

Thank you for bringing this up. We agree that the description of “closed” vs “open” states in our Arc207 dataset is a somewhat misleading description of the structure, as the “closed” structure is indeed a 5-mer “base” with an unresolved “lid”. Our 2D classification supports this notion – Figure 6B shows a faint density that is consistent with two dimers and DNA in the side-view classes. Unfortunately, neither additional 2D classifications nor expanding the number of requested classes (all done in the past two weeks) allowed better resolution of this faint density. However, additional 3D classifications allowed us to resolve more density. While the quality of this density is quite poor, due to a drastically reduced particle count, its presence supports the conclusion that the same deflected wrapping pathway consists for both states. To illustrate this, an additional supplemental video (Video 10) and supplemental figure (Figure 6—figure supplement 2) have been added to the manuscript. We have also modified the manuscript to no longer refer to “open” and “closed” conformations, but rather use the neutral terms “Class I” and “Class II” for the former closed and open forms

“A total of 1,879,294 particles were identified according to a neural network trained on manual particle selections and then classified. The two-dimensional classes of this dataset identifies two different populations of Arc207: one in which the classical “nucleosome-like” fold is observed (Figure 6B), and another in which two perpendicular DNA wrappings are clearly observed (Figure 6C).”

“To confirm that the structures described by the EM densities also exist in solution, and to correlate these findings with our SV-AUC analyses of particle compaction, we utilized the SoMo plugin of UltraScan to calculate sedimentation properties from our derived models. For the fully resolved Arc207 model, with the Arc90 “lid” bending ~90^o^ from the Arc120 “core”, we calculate a frictional ratio of 1.53, in close agreement with the 1.55 value extracted from the experimental traces. For the particles that can be fit with five histone dimers and 150 bp DNA (Class I), we surmised that the remaining 60 base pairs of DNA (and two histone dimers) extend from the observed density but assume variable orientations with respect to the main particle resulting in very weak density. Close inspection of the two-dimensional classes in this category indeed shows additional out-of-plane density that is consistent with the missing two dimers and ~60 base pairs of DNA, albeit at low contrast relative to background (Figure 6B). Further three-dimensional classifications of the 9.5 Å density reduced the overall resolution but yielded indications of the missing volume, albeit at low fidelity (Figure 6—figure supplement 2 ). To further model this particle, we extended the Arc150 structure with an additional Arc60 subunit positioned 90^o^ out-of-plane, similar to the model derived for the fully observed Arc207 density. This model predicted a similar value for the frictional coefficient of 1.50. In contrast, models with continuous wrapping and no deflections in the DNA pathway (as observed in the crystal structure, but not populated significantly in the cryo EM images) yield a frictional coefficient of 1.36. These data together show that archaeasomes on an extended DNA fragment can be viewed as a distribution of archaeasome subunits that can open up to a ~90^o^ angle between them.”

"Close inspection of the "closed state" two-dimensional classes indeed shows additional out-of-plane density that is consistent with the missing two dimers and ~60 base pairs of DNA, albeit at low contrast relative to background (Figure 7C)"There is no Figure 7. This statement likely refers to Figure 6C.

This has been corrected.

"Archaeasome compaction can be stabilized with divalent cations. In absence of archaeal ATP-dependent chromatin remodeling factors (large machines that regulate chromatin access in eukaryotes), this architecture provides an alternative mechanism for compacting chromatin and adjusting genome accessibility"There are several examples of regulatory mechanisms for chromatin remodeling in eukaryotes that involve domains which bind to specific histone post-translational modifications as a way to target remodeling activity, making it a mostly deterministic process (presence of a PTM causes recruitment of a remodeler). It is difficult to envision that any regulation mechanism in archaea could emerge from random conformational changes in their chromatin only. Is it definitely established from whole-genome sequencing that archaea do not have chromatin remodelers? Or could they have them, but these proteins have yet to be identified? Could the compaction induced by divalent cations be the main regulatory mechanism in vivo? (in which case membrane ion channels would also indirectly act as chromatin remodelers on a genome-wide scale by regulating intracellular concentrations of these ions?). The authors should consider discussing these points to enrich the Discussion section and potentially strengthen the connection between the their biophysical results and archaeal genome biology.

To date, there has not been conclusive evidence that archaeal histones are post-translationallly modified to any significant extent, and furthermore there are no histone tails to be modified. It is quite possible that the absence of ATP-dependent remodelers in archaea may be due to unsampled sequences or limited homology to eukaryotic machinery. We have modified our closing paragraph to expand this conversation. Regarding membrane pumps working as “indirect remodelers”, this is altogether possible, but we are afraid to “over-reach” in the discussion if we bring this up, as it is not known that Mg^2+^ concentrations vary in archaea. However, we hope that our findings here might encourage cell biologists to investigate Mg^2+^-regulation within *T. kodakarensis* and other archaea. We modified the text as follows:

“In eukaryotes, navigation of polymerases through chromatin is assisted by histone chaperones and ATP-dependent remodeling factors (33–35), but no known homologs to these complexes have been identified as yet in archaea. In the absence of remodelers, the inherent dynamics of archaeasome-based chromatin may thereby allow limited access to chromatin by the sporadic (and possibly stochastic) appearance of near-zero linker DNA and Arc60 or Arc90 substates. On the other hand, it is altogether possible that genes encoding chromatin remodelers are yet to be found. Genes relevant to chromatin organization continue to be identified in archaea, such as the “SMC-like” coalescin in the histone-less archaeon *Sulfolobus* (36), and the access of these yet unindentified “hidden agents” would similarly be tuned by the dynamics of the archaeasome. The degree to which DNA rigidity, variant histones, or unknown histone- and DNA-binding proteins regulate chromatin accessibility is an exciting future area of research.”

Materials and methods section, cryoEM grid preparationThe glow discharge conditions should be indicated.

These have been updated in the Materials and methods section (40 mA for 45 seconds).

"a Gatan K3 camera at 29,000x magnification in non-super resolution mode"This mode can be called "counting mode".

We have altered our text to refer to this as “counting mode”, as suggested.

"dosage rate of ~1 e/Å, and 50 frames per micrograph stack"The dose rate's unit is e/Å2/s, unless the authors are referring to the dose per frame (in which case it should be clarified). In addition, the total dose accumulated over the entire exposure time should be indicated.

Indeed, we intended to state this value as 1 e^-^/Å2 per frame (total does of 50 e^-^/Å2), and we have included these values in the Materials and methods text.

When first mentioning the Widom 601 sequence, the authors should cite (Lowary and Widom, 1998).

We have added the suggested citation to our first mention of the Widom 601 147 bp sequence.

"Gel shift assays showed that full complex saturation occurs when DNA and histones are mixed at the previously reported stoichiometric limit"This statement should cite an adequate reference.

This citation has been added (Mattiroli et al., 2017).

Reviewer #2 (Recommendations for the authors):It would be nice if they could include the rationale to study dimer-dimer and dimer-DNA interactions by MD using 90 bp, 120 bp and 180 bp.

Indeed, we only justify the simulation of the Arc90 system in our original description of our MD methods. As for the choice of 120 bp and 180 bp, these systems provide two different data points, in terms of archaeasome size (and number of L1-L1 interactions, consequently). In theory, we could have simulated archaeasomes of a varying number of sizes, even beyond 200 bp. However, the efficiency of an MD simulation scales non-linearly with the total number of atoms, and we believe that the additional computational time required to simulate larger systems would not provide any further information than what would be provided by our Arc180 system with multiple L1-stacking interactions. We have updated our Materials and methods section in an attempt to concisely explain these system choices:

“The Arc90 system represents the “fundamental unit” of archaeasome-based chromatin, as it is the crystallographic unit of the solved structure as well as the smallest DNA protection footprint observed by MNase digestion (17). The Arc120 and Arc180 complexes provide systems in which to study the contribution of stacking histone-histone interactions, as well as the wrapping of additional DNA superhelical turns, in stabilizing or destabilizing the proposed archaeasome.”

A MNase ladder is mentioned, but no citation or figure is referenced, please add the MNase ladder.

We have included references to the manuscript that contains the MNase digestion assay (Mattiroli, et al., 2017).

The authors very nicely show that Mg^2+^ does not impact archaeasome oligomerization, how does salt concentration impact oligomerization?

While we can determine that Mg^2+^ did not cause precipitation at higher concentrations via our tabletop experiment, potential soluble aggregates due to Mg^2+^ or high monovalent salt concentrations would require additional AUC experiments and optimization, which we decided to forego in light of the editor’s verdict that no additional experiments are necessary.

The authors mention that particles that interacted with neighboring particles were not used in the analysis. We are curious whether these particles would more closely resemble what would happen in cells where this is higher density (one presumes) of chromatin?

This is a good point, but would require a substantial re-analysis of our data to conclusively show the effect of “crowding” (if any) on the reconstructed densities. Nevertheless, our future work on archaeasome structures will be sure to include a comparison analysis of datasets processed “with” and “without” neighboring particles to determine insight into the effect of crowding.

What was the distribution of the angle of the open conformation found for the archaeasome?

Because we were only able to construct one 3D density with a well-resolved “lid”, we cannot say what the distribution of the angle is. We attempted to determine a distribution using 3D variability analysis in cryoSPARC; however, this analysis was inconclusive as the number of particles was quite low (~6,000, as mentioned in the manuscript) and the major contributions to the variability between particles was different PCA modes of noise, rather than altered signal. Our identification of a similar 90^o^ out-of-plane density classified deep within what we now refer to as “Class I” suggest that they may exhibit appreciable dynamics (as shown by the largely smeared electron density), potentially depending on the relative sizes of the attached archaeasome subunits, but we unfortunately cannot (as of yet) state the range of angles available between them without overinterpreting the statistical power of our data.

It would be interesting and timely if the authors could discuss their results in light of the recent publication of the archaea genome organization by Takemata and Bell, 2020.

The concept of chromosome compartmentalization is indeed an intriguing one. However, the subject of this study was archaeons from the *Sulfolobus* genus, which are some of the few archaea that do not encode histones. While the sequence of the “coalescin” protein that they have characterized is present with high identity in many different branches in archaea (determined via a blast alignment that we conducted), the generality of this compartmentalization is not well understood outside of Sulfolobus. As a result, we cannot conclusively extrapolate the roles or restrictions of archaeasomes in the context of this compartmentalization, as it has not yet been reported in *T. kodakarensis* or other similar archaeons. Nevertheless, we believe that the identification of archaeal chromatin reorganizing proteins, such as coalescin, highlights the need to continue farming these sequences in search of other chromatin remodelers, and we have included a reference to this study in our closing discussion:

“On the other hand, it is possible that sequences encoding chromatin remodelers are yet to be found. On a related note, the “SMC-like” coalescin proteins was recently identified in the histone-less *Sulfolobus* archaea (36), and the access of these yet unidentified “hidden agents” would similarly be tuned by the dynamics of the archaeasome.”